# Patterns of opioid dose escalation in patients with chronic kidney disease initiated on opioids for the treatment of non-cancer pain

Che Suraya Zin[1,2]*, Stefania Lando[1], Ailema González-Ortiz[1,3], Viyaasan Mahalingasivam[1,4,5], Shayan Mostafaei[1], Wan Rohaidah Ahmad[6], Mazlila Meor Ahmad Shah[7], Björn Wettermark[8], Juan Jesus Carrero[1,9]*

1 Department of Medical Epidemiology and Biostatistics, Karolinska Institutet, Stockholm, Sweden, 2 Department of Pharmacy Practice, Kulliyyah of Pharmacy, International Islamic University Malaysia, Kuantan, Malaysia, 3 Translational Research Center, Instituto Nacional de Pediatria, Mexico City, 4 Department of Non-Communicable Disease Epidemiology, London School of Hygiene & Tropical Medicine, London, United Kingdom, 5 Department of Nephrology & Transplantation, Barts Health NHS Trust, London, United Kingdom, 6 Pain management Unit, Department of Anaesthesiology, and Intensive Care Hospital Sultanah Nur Zahirah, Kuala Terengganu, Terengganu, Malaysia, 7 Pain management Unit, Department of Anaesthesiology and Intensive Care, Hospital Selayang, Batu Caves, Selangor, Malaysia, 8 Department of Pharmacy, Faculty of Pharmacy, Uppsala University, Uppsala, Sweden, 9 Department of Medical Specialist Care, Nephrology Clinic, Danderyd University Hospital, Stockholm, Sweden

* juan.jesus.carrero@ki.se (JJC), chesuraya@iium.edu.my (CSZ)

## Abstract

### Background

Pain management in chronic kidney disease (CKD) is challenging due to altered drug metabolism, impaired excretion, and higher opioid toxicity risk. Despite this, opioids are commonly prescribed, yet real-world data on dose escalation in CKD remain limited.

### Objective

To investigate patterns and timing of opioid dose escalation to ≥50 and ≥90 MME/day among new opioid users across kidney function levels.

### Methods

This population-based cohort study used data from the Stockholm Creatinine Measurements (SCREAM) project linking diagnoses, prescriptions, and laboratory records. Adult new opioid users (no prior opioid in 12 months) from 2012–2021 were categorized by baseline eGFR (≥60, 30–59, <30 mL/min/1.73m²). Opioids were identified using ATC codes, and daily doses (MME/day) were calculated based on strength, quantity, and equianalgesic ratios. Fine–Gray competing-risks regression assessed time to dose escalation (≥50 and ≥90 MME/day), accounting for death as a competing event.

**Data availability statement:** The data used in this study originate from Swedish national healthcare registers and are protected under the General Data Protection Regulation (GDPR) and Swedish legislation. Therefore, the data cannot be shared publicly. Researchers interested in accessing Swedish register data may submit an application to the Swedish National Board of Health and Welfare (Socialstyrelsen) in accordance with Swedish law. Information on how to apply for data access is available from the Swedish National Board of Health and Welfare (Socialstyrelsen) website (https://www.socialstyrelsen.se/en/statistics-and-data/apply-for-data/) or via email (registerservice@socialstyrelsen.se).

**Funding:** The author(s) received no specific funding for this work.

**Competing interests:** This research received no specific grant from any funding agency in the public, commercial, or not-for-profit sectors. CSZ was supported by a research grant from The Ministry of Education Malaysia (Fundamental Research Grant Scheme, FRGS/1/2022/SKK16/UIAM/01/3). The funders were not involved in the design of the study and collection, analysis, and interpretation of data and in writing the manuscript.

## Results

Of 81,987 adult new opioid users, 5,987 (7.3%) escalated to ≥50 MME/day comprising 7.4%, 6.8%, and 8.1% of patients with eGFR ≥ 60, 30–59, and <30 mL/min/1.73m², respectively. For ≥90 MME/day, 2,067 (2.5%) escalated, 2.5%, 2.3%, and 2.9% across the same eGFR categories. Competing risks regression showed significantly lower risks of escalation among patients with reduced eGFR levels. For ≥50 MME/day, the sub distribution hazard ratios (SHRs) were 0.67 (95% CI: 0.56–0.81, $p < 0.001$) for eGFR 30–59 and 0.64 (95% CI: 0.42–0.99, $p = 0.043$) for those with eGFR < 30.For ≥90 MME/day, SHRs were 0.57 (95% CI: 0.43–0.75, $p < 0.001$) and 0.31(95% CI: 0.15–0.65, $p = 0.002$), respectively. Most escalation occurred within six months, with minimal increase thereafter.

## Conclusion

Opioid dose escalation occurred across all eGFR levels, underscoring the need for cautious, individualized prescribing and close monitoring, especially in patients with reduced kidney function.

## Introduction

Patients with chronic kidney disease (CKD) often suffer from chronic pain, including musculoskeletal, neuropathic, and bone-related pain associated with CKD–mineral and bone disorders, affecting up to 60% of patients, particularly in more advanced stages [1]. Managing pain in this population poses unique challenges due to altered pharmacokinetic profiles, heightened drug sensitivity, and the accumulation of active metabolites resulting from impaired renal clearance [2,3]. These physiological differences, along with the burden of polypharmacy and multimorbidity, contribute to an elevated risk of medication-related complications in CKD [4].

Opioid analgesics remain a mainstay in the management of chronic pain in CKD patients, with studies showing that over the course of a year, approximately half of this population receives at least one opioid prescription [3,5]. While opioids may be appropriate for selected cases, inappropriate dosing in people with impaired kidney function can result in serious adverse outcomes, including sedation, respiratory suppression, and fatal overdose [6]. There remains a lack of specific clinical guidelines for safe opioid prescribing in this vulnerable group [7].

To improve opioid safety in the general population, the U.S. Centers for Disease Control and Prevention (CDC) issued guidelines in 2016 discouraging routine dose increases beyond 50 milligram morphine equivalents per day (MME/day) and urging caution above 90 MME/day [6]. Studies in the general population have shown significantly increased overdose risks at these thresholds compared to lower doses [8,9]. While prior research has largely focused on extreme opioid doses (≥100 or ≥200 MME/day) in the general population [8,9], little is known about early escalation to ≥50 and ≥90 MME/day, thresholds that are clinically meaningful and embedded in safety

guidelines especially among CKD patients. It is not well studied if these recommendations are followed in persons with CKD or if overdose risks occur at lower doses in persons with altered drug clearance.

We designed this study using real-world electronic health record data to examine to what extent patients with CKD initiated on opioids get their dosage regimens changed in clinical practice. Our objectives were to describe the patterns of opioid dose escalation to ≥50 MME/day and ≥90 MME/day across eGFR categories and evaluating the timing of dose escalation following the initial opioid prescription among patients stratified by kidney function. We hypothesized that patients with normal kidney function (i.e., estimated glomerular filtration rate [eGFR] ≥60 mL/min/1.73m$^2$) would be more likely to escalate to higher opioid doses and at a faster rate than those with CKD, reflecting both clinical caution and pharmacologic limitations. Understanding these patterns is essential to support safer prescribing practices and develop more tailored opioid use strategies for patients with CKD.

## Materials and methods

### Data source

This observational study utilized data from the Stockholm Creatinine Measurements (SCREAM) project, a population-based database encompassing all individuals in the region of Stockholm, Sweden, who accessed healthcare services between 2012 and 2021 [10]. Laboratory data were linked with several regional and national health registries, including the Stockholm regional healthcare data warehouse (VAL), the Swedish Prescribed Drug Register (PDR), the Cause of Death Register, and the Swedish Renal Register (SRR), enabling comprehensive retrieval of information on demographics, clinical diagnoses, healthcare utilization, and dispensed prescription drugs. The PDR captures details on all prescriptions across Sweden since July 2005, including drug names (coded using the Anatomical Therapeutic Chemical [ATC] classification), dosage, prescription dates, dispensing dates, and details of the prescribing clinic [11].

### Ethics approval

This study was conducted in accordance with the Declaration of Helsinki and adhered to the STROBE guidelines. Ethical approval for the SCREAM project was originally granted by the Regional Ethics Review Board in Stockholm (reference 2017/793–31), with a subsequent amendment approved by the Swedish Ethical Review Authority (reference 2024-07754-02). The Swedish National Board of Health and Welfare approved the use of national health register data.

SCREAM data are pseudo-anonymized before release, and the research team received only fully de-identified datasets. Consequently, informed consent was not required according to Swedish law. The data used in this study were accessed for research purposes on 20 June 2023, and no identifiable information was accessible to the authors during or after data collection.

### Study population and study design

This was a cohort study following a new-user design. We identified all patients aged 18 years and above who received at least two opioid prescriptions between January 1, 2012, and December 31, 2021. Patients were required to have not received any opioids in the 12 months preceding their first opioid prescription to qualify as new opioid users (Fig 1) The first opioid prescription was considered the date of opioid initiation and the study's index date. If a patient had multiple episodes of opioid use during the follow-up period, only their first episode of opioid use was considered (S1 Fig). This study defined an episode as a period of continuous opioid use, beginning with the first prescription and continuing as long as there were no gaps of more than 60 days between consecutive prescriptions. The gap between dispensations was defined as the interval from the end of supply of the previous prescription to the date of the subsequent opioid prescription dispensed. This was considered the first treatment episode. If a gap exceeded 60 days, a new episode was assumed to begin with the next prescription. This approach has also been employed in previous studies. [12,13] We excluded anyone

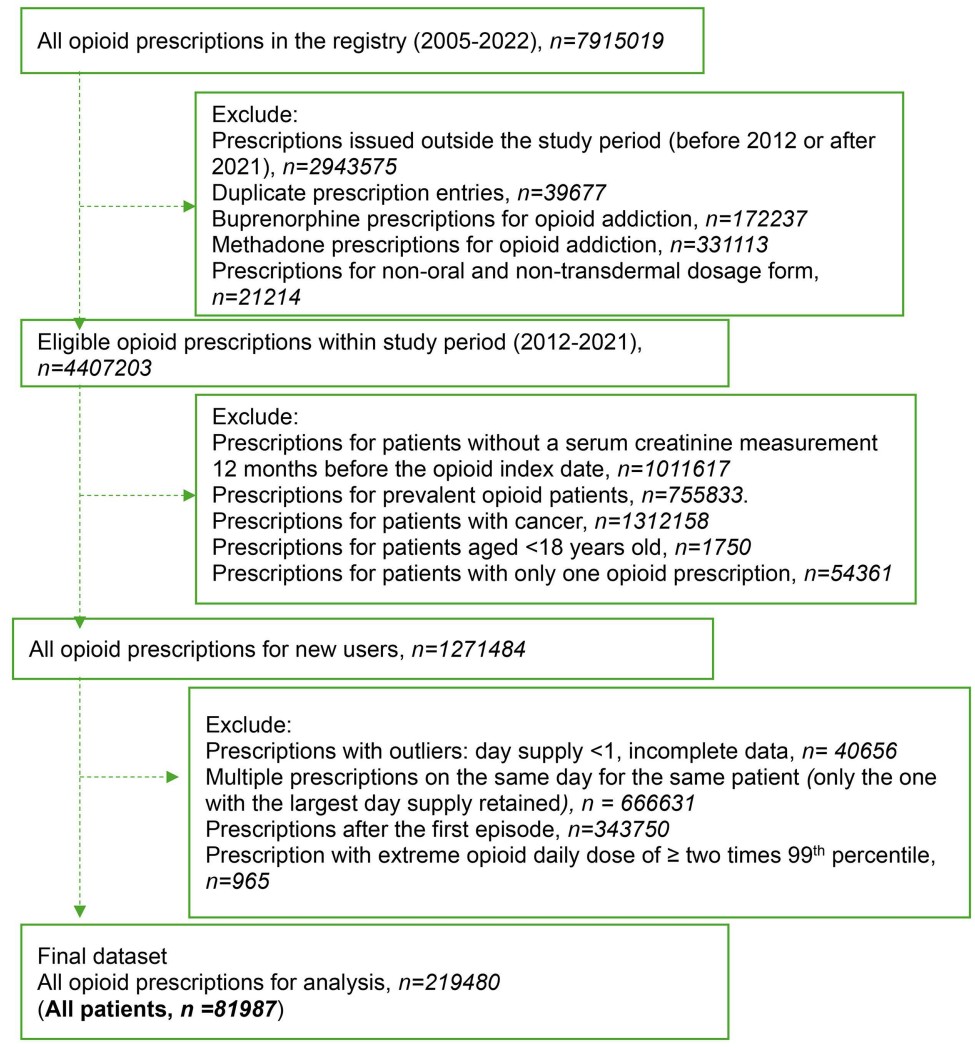

**Fig 1. Study cohort flowchart.**

without at least one outpatient measurement of serum/plasma creatinine at the time of opioid therapy initiation or up to 18 months before. Patients were followed until an event, death, disenrollment, or the end of the study (i.e., December 31, 2021), whichever comes first.

We evaluated the following opioids commercially available in Sweden; morphine (ATC: N02AA01), buprenorphine (N02AE01), oxycodone (N02AA05), fentanyl (N02AB03), tramadol (N02AX02), ketobemidone (N02AB01), and tapentadol (N02AX06) (S1 Table). These opioids were identified from The Swedish PDR. Two fixed combination products including codeine/paracetamol and codeine/acetylsalicylic acid, respectively, were excluded because of their low potency, limited escalation potential, variable metabolism, and formulation constraints.

To distinguish prescriptions for buprenorphine and methadone dispensed as medication-assisted treatment (MAT) for opioid use disorder (OUD) from those for analgesia, this study excluded the ATC codes assigned for MAT for buprenorphine (N07BC01) and methadone (N07BC02). Any prescriptions for non-oral and non-transdermal such as injectable or suppository opioids were also excluded. Since this study included dispensed prescriptions, hospital-administered opioids

were not included. No opioids are available without prescription as over the counter (OTC) in Sweden during the time of the study.

## Measurement of opioid dose

The period of opioid exposure started on the day the first prescription was dispensed and ended after the end of supply from the last prescription. The number of days that each opioid prescription covers was determined by dividing the quantity supplied by the daily frequency. The total number of days that a patient was covered by opioid medication was calculated by adding up the number of days covered by all of their prescriptions. Any overlapping days of opioid coverage between prescriptions during the study period were subtracted. For multiple opioid prescriptions on the same day for the same patient, only the prescription with the largest day supply was included. All dispensed opioid quantities were included in the total milligrams of morphine equivalent (MME) calculations; overlaps only affected the estimation of days' supply, where stockpiling rules were applied to avoid double counting. [14].

The total morphine equivalents for each prescription during first episode after the index date was calculated by multiplying the quantity of each prescription by the strength in milligrams of opioid per unit dispensed of the prescription. This quantity–strength was then multiplied by equianalgesic ratio of the opioid to derive the MME of the prescription (S1 Table). The individual monthly total opioid dose for each patient was obtained by summing the opioid dose across all prescriptions for a particular patient during the follow-up period after initiating opioid. The mean daily opioid dose in MME was calculated by dividing the total opioid dose by the total number of days covered with opioid for each patient. Prescriptions with extreme values such as days' supply or doses exceeding two times 99th percentile MME were excluded from the study.

## Estimation of glomerular filtration rate (eGFR)

In clinical practice, kidney function is categorised using eGFR [15]. A sustained reduction in eGFR below 60 mL/min/1.73m², is indicative of chronic kidney disease (CKD) [15]. We calculated baseline eGFR using the 2021Chronic Kidney Disease Epidemiology Collaboration (CKD-EPI) equation [16] from the outpatient serum/plasma creatinine value closest to the index opioid prescription date and no longer than 18 months prior. Adjustment for race was not applied. All creatinine assays were calibrated to isotope-dilution mass spectrometry standards, in accordance with manufacturer protocols.

## Exposure

The primary exposure was baseline kidney function categorised as no CKD or CKD stages G1/G2 (mild CKD) (eGFR ≥ 60 mL/min/1.73m²), CKD stage G3 (moderate CKD) (eGFR 30–59 mL/min/1.73m²), and CKD stages G4/G5 (advanced CKD) (<30 mL/min/1.73m²) [15].

## Outcomes

The primary outcome was opioid dose escalation to ≥50 MME/day. This included the number of patients who experienced escalation and the time to escalation from initial prescription. A secondary outcome assessed escalation to ≥90 MME/day, a threshold linked to heightened risk of opioid-related adverse events [6]. To ensure outcomes reflected initial escalation rather than repeated episodes, only the first event of dose escalation was considered for each patient.

## Covariates

Information on age, and sex were extracted at the index date. Comorbidities were defined through issued diagnoses (S2 Table) recorded in inpatient, outpatient specialist or primary care. We included only comorbidities with direct clinical relevance to opioid prescribing and dose escalation risk. These included chronic or acute pain, history of use disorders

(alcohol use disorder, other substance use disorder, cannabis use disorder, and tobacco use disorder), and mental health disorders (anxiety disorder, bipolar disorder, depressive disorder, and schizophrenia disorder). These comorbidities were identified within one year prior to the index date, a timeframe chosen to ensure temporal relevance to opioid use, which is typically recommended for short-term use. Furthermore, in Sweden, prescriptions are valid for a maximum of one year, and patients with pain or other chronic conditions relevant to this study typically have at least annual follow-up, which supports the choice of a 1-year look-back window.

Comedications were measured as dispensed prescriptions within 6 months before the index date. The following comedications were included in this study; benzodiazepines and Z-hypnotics, non-steroidal anti-inflammatory drugs (NSAIDs), antidepressants, antipsychotics and NSAIDs (S3 Table).

### Statistical analysis

Patient characteristics were summarized within each eGFR category using descriptive statistics. Continuous variables were reported as median (interquartile range, IQR), and categorical variables were expressed as counts and percentages. To assess time to opioid dose escalation at thresholds of ≥50 MME/day and ≥90 MME/day, we first employed Kaplan–Meier curves to graphically depict the probability of remaining below the respective dose threshold over time. Log-rank tests were used to evaluate differences in time to escalation between eGFR categories. Next, we used multivariable Cox proportional hazards regression to estimate cause-specific hazard ratios (HRs) for time to dose escalation, using eGFR ≥ 60 mL/min as the referent category. The models were adjusted for all identified confounders (see section of covariates). Interaction terms were also tested (e.g., between psychiatric conditions and their treatments, and between sex and age) to assess potential effect modification. The proportional hazards assumption was tested by using Schoenfeld residuals, including assessments of time-dependency. Model fit was further evaluated using Martingale and Cox–Snell residuals.

As death represents a frequent competing event in this population, Kaplan–Meier and Cox models may overestimate the cumulative risk of dose escalation. Therefore, these analyses were used primarily for unadjusted descriptive purposes, while the Fine–Gray competing-risks regression was applied as the main analytical approach to provide more accurate risk estimates.

We also employed Fine–Gray competing-risks regression to model time to opioid dose escalation to ≥50 MME/day and ≥90 MME/day, treating death as the competing event and reporting sub distribution hazard ratios (SHRs) with 95% CIs. Standard errors were cluster-robust at the patient level. Model-based cumulative incidence functions (CIFs) by eGFR category were obtained from the Fine–Gray model (other covariates held at their sample means) and summarized at prespecified time points (3, 6, 9, 12 months; 2, 3, 5, 8 years). As an unadjusted check, we also estimated Aalen–Johansen CIFs by eGFR and performed a Gray-equivalent global test using an unadjusted Fine–Gray model.

To test the stability of the findings, a sensitivity analysis was performed by excluding patients with eGFR values above 105 mL/min/1.73 m$^2$. These elevated eGFR values could represent hyperfiltration cases or, inaccurate GFR estimation in individuals with low muscle mass, multimorbidity or acute illness. All analyses were conducted using Stata version 19 [17]. Statistical tests were two-sided, and a p-value below 0.05 was considered statistically significant.

## Results

### Patient characteristics

A total of 219,480 opioid prescriptions were dispensed to 81,987 unique patients during the study period (Fig 1). Of these patients, 70,199 (85.6%) had eGFR ≥ 60 mL/min/1.73m², 10,395 (12.7%) had eGFR 30–59 mL/min/1.73m², and 1,393 (1.7%) had eGFR < 30 mL/min/1.73m² (Table 1). Median age was higher among patients with lower eGFR categories. The proportion of female patients increased with lower eGFR, from 55.8% in the eGFR ≥ 60 group to 68.1% and 58.7% in the eGFR 30–59 and eGFR < 30 groups, respectively.

**Table 1. Patient demographics.**

| Descriptions | Overall | eGFR ≥ 60 mL/min/1.73m² | 30 ≤ eGFR < 60 mL/min/1.73m² | eGFR < 30 mL/min/1.73m² |
|---|---|---|---|---|
| Total number of patients, n (%) | 81987 | 70199(85.6) | 10395(12.7) | 1393(1.7) |
| Age, years, median (IQR) | 63(48, 77) | 59 (45, 72) | 83 (75, 89) | 83 (71, 90) |
| Age group, n (%) | | | | |
| <=40 years | 13,828 (16.9%) | 13716 (19.5%) | 61 (0.6%) | 51 (3.7%) |
| 41-50 years | 9,844 (12.0%) | 9679 (13.8%) | 116 (1.1%) | 49 (3.5%) |
| 51-60 years | 13,745 (16.8%) | 13291 (18.9%) | 366 (3.5%) | 88 (6.3%) |
| 61-80 years | 28,958 (35.3%) | 24909 (35.5%) | 3639 (35.0%) | 410 (29.4%) |
| >80 years | 15,612 (19.0%) | 8604 (12.3%) | 6213 (59.8%) | 795 (57.1%) |
| Sex, n (%) | | | | |
| Male | 34,936 (42.6%) | 31048 (44.2%) | 3312 (31.9%) | 576 (41.3%) |
| Female | 47,051 (57.4%) | 39151 (55.8%) | 7083 (68.1%) | 817 (58.7%) |
| Laboratory values | | | | |
| Baseline eGFR, median (IQR)mL/min/1.73m2 | 87.9 (71.4, 101.7) | 91 (79, 104) | 49 (42, 55) | 23 (16, 27) |
| Diagnoses n (%) | | | | |
| Chronic or acute pain diagnosis | 31,289 (45.8%) | 27501 (44.6%) | 3347 (56.1%) | 441 (56.6%) |
| Alcohol use disorder | 3,696(5.40%) | 3484 (5.7%) | 190 (3.2%) | 22 (2.8%) |
| Opioid use disorder | 1,259(1.84%) | 1190 (1.9%) | 53 (0.9%) | 16 (2.1%) |
| Other substance use disorder | 2,016(2.95%) | 1948 (3.2%) | 56 (0.9%) | 12 (1.5%) |
| Cannabis use disorder | 399(0.58%) | 396 (0.6%) | 1 (<1%) | 2 (0.3%) |
| Tobacco use | 1,461(2.14%) | 1347 (2.2%) | 98 (1.6%) | 16 (2.1%) |
| Anxiety disorder | 15,448(22.6%) | 14216 (23.1%) | 1098 (18.4%) | 134 (17.2%) |
| Bipolar disorder | 954(1.39%) | 915 (1.5%) | 29 (0.5%) | 10 (1.3%) |
| Depressive disorder | 11,283(16.5%) | 10147 (16.5%) | 1020 (17.1%) | 116 (14.9%) |
| Schizophrenia spectrum disorder | 585(0.86%) | 498 (0.8%) | 77 (1.3%) | 10 (1.3%) |
| Comedications, n (%) | | | | |
| Anti-depressants | 16983(20.7%) | 14402 (20.5%) | 2290 (22.0%) | 291 (20.9%) |
| Anti-psychotic | 6075 (7.4%) | 4919 (7.0%) | 1027 (9.9%) | 129 (9.3%) |
| Benzodiazepines or Z hypnotics | 12701(15.5%) | 10173 (14.5%) | 2241 (21.6%) | 287 (20.6%) |
| Gabapentinoids | 13108(16.0%) | 11574 (16.5%) | 1380 (13.3%) | 154 (11.1%) |
| NSAIDs | 18917(23.1%) | 17541 (25.0%) | 1286 (12.4%) | 90 (6.5%) |

Pain-related diagnoses were common in all eGFR categories, reported in 44.6% of patients with eGFR ≥ 60, 56.1% in those with eGFR 30–59, and 56.6% in those with eGFR < 30. The prevalence of anxiety disorders was lower across lower eGFR, from 23.1% in the eGFR ≥ 60 group, to 17.2% in the eGFR < 30 group. The prevalence of depressive disorders was similar across the three groups, reported in 16.5%, 17.1%, and 14.9%, respectively. Use antidepressant medications was similar across the three categories. However, the use of antipsychotics, and benzodiazepines or Z-hypnotics was higher with lower eGFR. In contrast, the use of NSAIDs and gabapentinoids was highest among patients with eGFR ≥ 60 and lowest among those with eGFR < 30 ml/min/1.73m².

### Opioid dose escalation to 50 MME/day

A total of 5,987(7.3%) patients experienced opioid dose escalation to ≥50 MME/day during a median follow-up of 14 months (IQR 12.5–37.3). This included 5170 (7.4%) patients in the eGFR ≥ 60 ml/min/1.73m² group, 704 (6.8%) in the eGFR 30–59 group, and 113 (8.1%) in the eGFR < 30 group. KM curves showed no between-group difference in time to

escalation ≥50 MME/day (log-rank χ²=2.28, p=0.32) and adjusted Cox likewise found no association with eGFR (S4 Table).

As no significant differences were observed using Kaplan–Meier or Cox models, subsequent analyses focused on the Fine–Gray competing-risks regression, which appropriately accounts for death as a competing event.

However, when using a Fine-Gray competing risks regression model that accounted for death as a competing event, kidney function was significantly associated with the likelihood of dose escalation (Fig 2). Compared to patients with eGFR≥60 mL/min, those with eGFR 30–59 had a 33% lower sub distribution hazard of escalation (SHR 0.67, 95% CI: 0.56–0.81, p<0.001), while those with eGFR<30 had a 36% lower sub-hazard (SHR 0.64, 95% CI: 0.42–0.99, p=0.043) (S5 Table). Adjusted time-point cumulative incidence was highest for eGFR≥60 at every horizon: 3/6/9/12 months=8.6/16.1/19.9/22.2% (≥60 mL/min) vs 5.9/11.2/13.9/15.6% (30–59) vs 5.6/10.6/13.2/14.8% (<30); 2/3/5/8 years=25.7/27.0/28.4/29.6% (≥60) vs 18.2/19.1/20.2/21.1% (30–59) vs 17.3/18.3/19.2/20.1% (<30) (S6 Table). The unadjusted Aalen–Johansen CIFs differed significantly across eGFR groups (Gray's test: χ²(2)=264.8, p<0.001), consistent with the adjusted Fine–Gray results.

In Fine–Gray competing-risks models adjusted for identified confounders, several factors were associated with a higher sub-hazard of opioid dose escalation to ≥50 MME/day: gabapentinoid use (SHR 2.13, 95% CI 1.91–2.38; p<0.001), antidepressant use (SHR 1.22, 95% CI: 1.06–1.39, p=0.005), NSAID use (SHR 2.07, 1.79–2.40; p<0.001), pain-related diagnoses (SHR 2.47, 1.86–3.28; p<0.001), opioid use disorder (SHR 1.58, 1.30–1.93; p<0.001), anxiety disorders (SHR 1.22, 1.08–1.39; p=0.002), and bipolar disorder (SHR 1.47, 1.14–1.90; p=0.003). In contrast, antipsychotic use (SHR 0.76, 0.66–0.88; p<0.001), benzodiazepine use (SHR 0.89, 0.79–0.99; p=0.037), alcohol use disorder (SHR 0.83, 0.70–0.98; p=0.032), and older age (per-year SHR 0.97, 0.964–0.970; p<0.001) were associated with lower sub-hazards; female gender was not significant (SHR 0.95, 0.86–1.06; p=0.38). No significant associations were also observed for other confounders such as schizophrenia and cannabis use disorder (p≥0.05) (S5 Table)

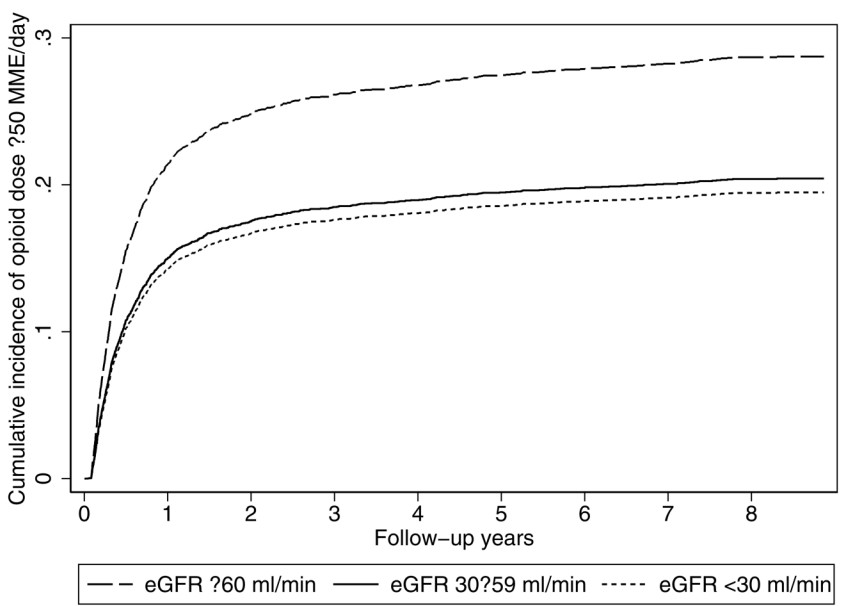

**Fig 2. Model-based cumulative incidence (Fine–Gray): escalation to ≥50 MME/day by eGFR category (death as competing event).**

## Opioid dose escalation to 90 MME/day

A total of 2,067(2.5%) patients experienced opioid dose escalation to ≥90 MME/day during a median follow-up of 14 months (IQR 10–28). This included 1,785 (2.5%) patients with eGFR ≥ 60 ml/min, 241 (2.3%) patients with eGFR 30–59, and 41(2.9%) patients with eGFR < 30.KM curves showed no between-group difference for escalation ≥90 MME/day (log-rank χ²=0.31, p=0.854). In adjusted Cox, eGFR 30–59 mL/min had a modestly higher hazard (HR 1.21, 95% CI 1.03–1.43, *p*=0.019), while eGFR < 30 was not significant (S7 Table).

As no significant differences were observed using Kaplan–Meier or Cox models, subsequent analyses focused on the Fine–Gray competing-risks regression, which appropriately accounts for death as a competing event.

In Fine–Gray competing risks regression accounting for death, both groups with reduced eGFR levels had significantly lower sub distribution hazards of escalation ≥90 MME/day compared to patients with eGFR ≥ 60 mL/min: patients with eGFR 30–59 had a 43% lower risk (SHR=0.57, 95% CI: 0.43–0.75, *p<0.001*), and those with eGFR < 30 had a 69% lower risk (SHR=0.31, 95% CI: 0.15–0.65, *p=0.002*) (Fig 3) (S8 Table). Adjusted time-point CIFs were highest for eGFR ≥ 60, intermediate for 30–59, and lowest for <30 at every horizon: 3/6/9/12 months = 3.5/7.6/10.9/12.6% (≥60) vs 2.0/4.4/6.4/7.4% (30–59) vs 1.1/2.4/3.5/4.1% (<30). At 2/3/5/8 years, risks were 17.7/19.4/20.5/21.1% (≥60) vs 10.5/11.6/12.3/12.6% (30–59) vs 5.9/6.5/6.9/7.1% (<30) (S9 Table). The unadjusted Aalen–Johansen CIFs for ≥90 MME/day differed significantly across eGFR groups (Gray's test: χ²(2)=192.5, p<0.001), consistent with the adjusted Fine–Gray results.

Concerning identified confounders, adjusted Fine–Gray competing-risks models showed that several covariates were associated with a higher sub distribution hazard of dose escalation ≥90 MME/day: pain-related diagnoses (SHR 2.66, 95% CI 1.79–3.97; *p<0.001*), opioid use disorder (1.97, 1.55–2.50; *p<0.001*), anxiety disorders (1.40, 1.18–1.67; *p<0.001*), antidepressant use (1.38, 1.13–1.69; *p=0.002*), gabapentinoid use (2.42, 2.08–2.83; *p<0.001*), and NSAID use (2.11, 1.71–2.60; *p<0.001*). In contrast, antipsychotic use (0.70, 0.57–0.85; *p<0.001*), alcohol use disorder (0.68, 0.54–0.86; *p=0.001*), cannabis use disorder (0.53, 0.29–0.95; *p=0.034*), and older age (0.95, 95% CI 0.949–0.958;

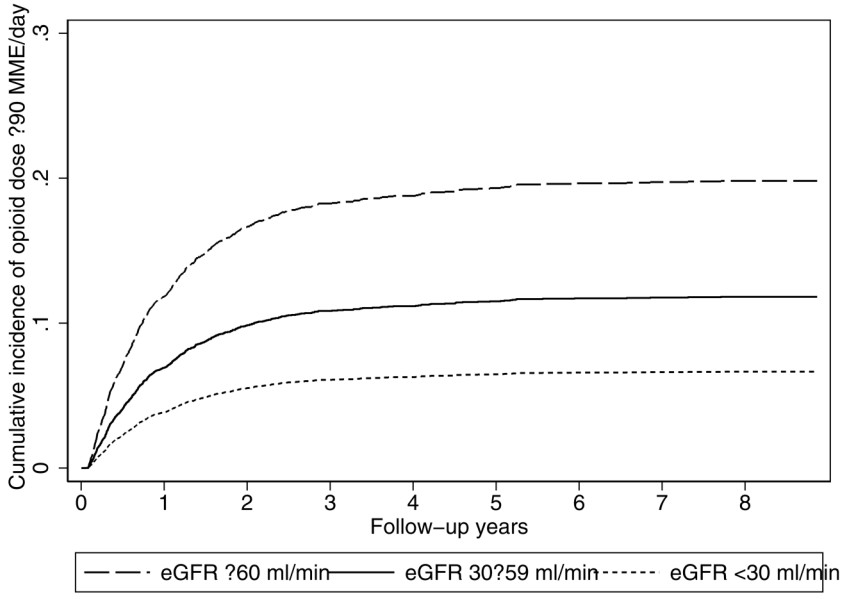

**Fig 3. Model-based cumulative incidence (Fine–Gray): escalation to ≥90 MME/day by eGFR category (death as competing event).**

*p<0.001*)(S8 Table) were associated with lower sub distribution hazards. Other confounders such as schizophrenia and substance-use, benzodiazepine use, and female sex were not statistically significant.

## Sensitivity analyses

In sensitivity analyses, exclusion of patients with eGFR > 105 mL/min/1.73 m² did not materially alter the findings, indicating that the main results were robust to potential misclassification in this subgroup.

## Discussion

This large population-based study provides insights into patterns of opioid dose escalation in clinical practice by varying degrees of kidney function. We found that patients with CKD had lower risks of opioid dose escalation across both thresholds (≥50 and ≥90 MME/day). The pattern was consistent at all prespecified time points, with escalation concentrated in the first year particularly by 6 months and only modest accrual thereafter.

In competing risks analysis, opioid dose escalation was less likely in patients with reduced kidney function (eGFR 30–59 and eGFR < 30mL/min). These findings align with prior literature emphasizing cautious opioid use in CKD. Conservative opioid prescribing in CKD is recommended due to altered drug clearance and increased toxicity risk [4,7]. Prior work has documented high variability in opioid use among CKD patients that often attributed to prescriber hesitation [3] and lower opioid dosing and reduced long-term use in patients with CKD, consistent with our observation of less frequent dose escalation in this population.

The timing of opioid dose escalation in the current study highlights a critical early window in which upward titration occurs. At 6 months, the adjusted cumulative incidence of escalation was 16.1%/11.2%/10.6% for ≥50 MME/day and 7.6%/4.4%/2.4% for ≥90 MME/day in the eGFR ≥ 60/30–59/< 30 mL/min groups, respectively. A study in chronic non-cancer pain patients observed escalation to the much higher threshold of ≥200 MME/day occurring over a median of 186 days (approximately 6.1 months) [18]. This suggests that escalation to ≥50 or ≥90 MME/day in our cohort occurred over comparable or even longer periods, despite representing significantly lower dose thresholds. The slightly longer escalation times in our cohort are expected, as patients with impaired kidney function typically require slower titration due to their increased sensitivity to opioids and elevated risk of adverse drug effects. These findings underscore the importance of early monitoring, especially in the first 4–8 months after opioid initiation, when the risk of dose escalation appears to be highest. This early period represents a crucial opportunity for clinicians to reassess pain control strategies and reinforce safer prescribing practices, particularly in populations at elevated risk of harm.

Pain in patients with CKD is heterogeneous and may arise from multiple mechanisms, including musculoskeletal conditions, neuropathic processes, and complications related to CKD-mineral and bone disorders [1,5,19]. These distinct pain phenotypes often require individualized management strategies. In addition to opioids, non-opioid pharmacological therapies such as paracetamol and gabapentinoids are commonly used in CKD, although their dosing and safety profiles require careful consideration in the context of impaired renal function [1,5]. Furthermore, non-pharmacological approaches, including cognitive-behavioral therapy, exercise-based interventions, and multidisciplinary pain management programs, play an important role in optimizing pain control while minimizing pharmacological risks.

Nutritional factors may also contribute to pain experiences in CKD, particularly in relation to mineral and bone disorders [19]. Dietary phosphorus restriction, which is routinely recommended in advanced CKD, has been proposed as a potential modifier of musculoskeletal and bone-related pain, although empirical evidence remains limited [19,20]. Future studies integrating clinical, nutritional, and patient-reported outcome data may help clarify the role of such interventions in comprehensive pain management strategies for this population.

Nevertheless, even within a context of recommended cautious and multimodal pain management, escalation to high-risk opioid doses was still observed in our study. While direct comparisons with previous studies are limited due to differences in population and dose thresholds, most existing literature focuses on extreme escalation, such as ≥100 or ≥200

MME/day. For instance, a study in patients with chronic non-cancer pain reported that 589 patients (1.8%) escalated to ≥200 MME/day [18], while Zin et al. (2019) found that 248 patients (5.3%) reached ≥100 MME/day and 69 (1.5%) exceeded ≥200 MME/day [21]. The lower proportion of patients in the current study reaching ≥90 MME/day (2.5%) likely reflects a more cautious prescribing approach in CKD.

These escalation thresholds are clinically important, given the well-documented dose-dependent risks associated with opioids. Bohnert et al. (2011) found that patients prescribed 50–99 MME/day had a 3.7-fold increased risk of opioid overdose death, rising to 8.9-fold at ≥100 MME/day [9]. Similarly, Gomes et al. (2011) reported a 1.9-fold increased risk at 50–99 MME/day and a 2.04-fold increase at ≥100 MME/day [8]. These findings reinforce the rationale behind the CDC guidelines, which discourage exceeding 50 MME/day and recommend extreme caution beyond 90 MME/day especially in high-risk populations such as those with impaired kidney function [6].

Although the Swedish national treatment recommends that opioid prescribing should be individualized and guided by a comprehensive patient assessment rather than fixed numerical dose thresholds [22], the CDC cut-offs of ≥50 and ≥90 MME/day are widely cited in the scientific literature and serve as internationally recognized risk indicators, enabling comparability across studies. These thresholds are also endorsed by the European Pain Federation (EFIC) as reference points for cautious opioid prescribing [23].

Several covariates were significantly associated with opioid dose escalation, consistent with prior studies. Use of gabapentinoids, NSAIDs, antidepressants, pain-related diagnoses, opioid use disorder and anxiety disorders likely reflects more complex clinical needs, which may predispose patients to higher opioid requirements and greater overdose risk [8,9,24,25]. These findings reinforce the importance of closely monitoring patients with such risk profiles.

We do not interpret the inverse associations (e.g., antipsychotics, benzodiazepines, alcohol/cannabis use, older age) as protective effects. Rather, they reflect selection into more conservative treatment pathways (cautious titration, deprescribing, or alternative therapies) and higher competing mortality, which together reduce the cumulative incidence of escalation in Fine–Gray analyses. Notably, literature linking benzodiazepines to overdose risk addresses a different outcome; our endpoint is dose escalation, so inverse or null associations here are compatible with elevated overdose risk reported elsewhere. Overall, these findings support tailored monitoring in patients with markers of more complex pain management while cautioning against causal interpretation of medication covariates in observational competing-risk analyses.

## Strengths and limitations

One of the key strengths of this study lies in its novel contribution to the evidence base, as it is among the first to comprehensively examine individual patient level data of opioid dose escalation patterns in patients with different eGFR categories. While prior studies have largely focused on high-dose thresholds (e.g., ≥ 100 or ≥200 MME/day), no studies to date have assessed dose escalation to ≥50 MME/day and ≥90 MME/day, thresholds that are clinically significant and directly aligned with CDC guidelines. This distinction is particularly important in a CKD population, given the altered drug metabolism, heightened toxicity risk, and unique prescribing challenges in this population. Another key strength of this study is the use of prescribed opioid dose, which reflects actual clinical practice. Unlike defined daily dose (DDD) or package's DDD (FORPDDD), it accounts for the specific dosing interval and duration prescribed, providing a more accurate estimate of opioid exposure.

The study also benefits from a large, population-based cohort, leveraging linked healthcare and prescription registries to capture high-quality real-world data on opioid prescribing across a full spectrum of kidney function. The long follow-up period and complete prescription records enabled precise estimation of opioid dosing trajectories and robust adjustment for a wide array of covariates, including comorbidities and co-medications. Additionally, the application of competing risks regression accounting for death as a competing event adds methodological rigor, particularly relevant in CKD where mortality risk is elevated.

Nonetheless, several limitations must be considered. First, reliance on prescription data does not confirm actual medication use. Second, data on over-the-counter medications, non-pharmacologic pain interventions, and pain severity were unavailable. Although no opioids are available OTC in Sweden, the lack of data on other analgesics including paracetamol could still influence both prescribing behaviour and dose escalation. Third, pain and mental-health diagnoses are under-recorded in administrative data; despite a 1-year ascertainment window before initiation, some conditions were likely underestimated as noted in prior Swedish registry work on pregabalin use [26], leaving scope for residual confounding. Fourth, codeine was not captured; patients may switch between codeine and other opioids. In Sweden, codeine is mainly available in fixed-dose combinations with paracetamol or aspirin, so the impact on our escalation estimates is likely limited. Additionally, kidney function was assessed at baseline at the time of opioid initiation. Changes in renal function over time may have influenced opioid dose escalation, which was not captured in this study. Finally, the relatively small number of patients with eGFR < 30 mL/min may reduce statistical power and limit the generalizability of findings for those with advanced CKD. Overall, these limitations would tend to dilute rather than create effects. The consistent patterns across both thresholds, the early clustering of risk in the CIFs, and robust sensitivity analyses indicate that our central finding on the lower cumulative incidence of escalation with reduced eGFR when accounting for competing death is unlikely to be materially affected.

## Conclusion

Our findings indicate that while patients with reduced kidney function had a significantly lower adjusted risk of escalating to high opioid doses, dose escalation still occurred across all levels of eGFR including among those with eGFR < 30 mL/min highlighting the need for cautious and individualized prescribing in this vulnerable population. These results address a critical gap in pain management, where clinical guidelines for opioid use in CKD remain lacking. By highlighting patterns of escalation and identifying key risk factors, our study offers valuable insight to support safer, evidence-informed prescribing decisions. In light of the well-documented risks associated with high-dose opioid use including overdose and mortality, it is imperative that prescribers select the lowest effective dose, tailored to each patient's renal function and clinical complexity.

Looking ahead, future research should evaluate whether opioid doses are being adequately adjusted in response to worsening kidney function and whether current prescribing practices align with patients' comorbidity burden. Developing renal-specific opioid prescribing frameworks could greatly enhance the safety and effectiveness of pain management in CKD ultimately improving outcomes for a growing and vulnerable patient population.

## Supporting information

**S1 Fig. Study cohort flowchart.**
(DOCX)

**S1 Table. ATC codes for opioids and opioid equianalgesic doses.**
(DOCX)

**S2 Table. ICD-10 codes for comorbidities and cancer diagnoses.**
(DOCX)

**S3 Table. ATC codes for comedications.**
(DOCX)

**S4 Table. Adjusted hazard ratios for dose escalation to ≥50 MME/day (Cox proportional hazards regression).**
(DOCX)

**S5 Table. Adjusted sub-hazard ratios for dose escalation to ≥50 MME/day (Fine–Gray competing risks regression).**
(DOCX)

**S6 Table. Adjusted cumulative incidence of opioid dose escalation to ≥50 MME/day at prespecified time points (Fine–Gray, death as competing event).**
(DOCX)

**S7 Table. Adjusted hazard ratios for dose escalation to ≥90 MME/day (Cox proportional hazards regression).**
(DOCX)

**S8 Table. Adjusted sub-hazard ratios for dose escalation to ≥90 MME/day (Fine–Gray competing risks regression).**
(DOCX)

**S9 Table. Adjusted cumulative incidence of opioid dose escalation to ≥90 MME/day at prespecified time points (Fine–Gray, death as competing event).**
(DOCX)

## Author contributions

**Conceptualization:** Che Suraya Zin, Björn Wettermark, Juan Jesus Carrero.

**Data curation:** Che Suraya Zin, Stefania Lando, Ailema González-Ortiz, Viyaasan Mahalingasivam, Shayan Mostafaei.

**Formal analysis:** Che Suraya Zin, Shayan Mostafaei, Björn Wettermark, Juan Jesus Carrero.

**Funding acquisition:** Che Suraya Zin, Juan Jesus Carrero.

**Investigation:** Che Suraya Zin, Wan Rohaidah Ahmad, Mazlila Meor Ahmad Shah, Björn Wettermark, Juan Jesus Carrero.

**Methodology:** Che Suraya Zin, Wan Rohaidah Ahmad, Mazlila Meor Ahmad Shah, Björn Wettermark, Juan Jesus Carrero.

**Project administration:** Che Suraya Zin, Stefania Lando, Ailema González-Ortiz, Viyaasan Mahalingasivam, Shayan Mostafaei.

**Resources:** Che Suraya Zin, Juan Jesus Carrero.

**Software:** Che Suraya Zin, Stefania Lando, Ailema González-Ortiz, Viyaasan Mahalingasivam, Shayan Mostafaei.

**Supervision:** Björn Wettermark, Juan Jesus Carrero.

**Validation:** Che Suraya Zin, Björn Wettermark, Juan Jesus Carrero.

**Visualization:** Che Suraya Zin, Stefania Lando, Wan Rohaidah Ahmad, Mazlila Meor Ahmad Shah, Björn Wettermark, Juan Jesus Carrero.

**Writing – original draft:** Che Suraya Zin.

**Writing – review & editing:** Che Suraya Zin, Stefania Lando, Ailema González-Ortiz, Viyaasan Mahalingasivam, Shayan Mostafaei, Wan Rohaidah Ahmad, Mazlila Meor Ahmad Shah, Björn Wettermark, Juan Jesus Carrero.

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
