## [Decision Letter · Decision Letter 0]

9 Feb 2026

Dear Dr. Carrero,

Thank you for submitting your manuscript to PLOS ONE. After careful consideration, we feel that it has merit but does not fully meet PLOS ONE’s publication criteria as it currently stands. Therefore, we invite you to submit a revised version of the manuscript that addresses the points raised during the review process.

Clinical guidelines are lacking for opioid use in patients with chronic kidney disease (CKD), who have higher risks for overdose and mortality from high-dose opioid use. The authors' study found that patient data stratified by eGFR generally showed a lower risk of escalation to higher opioid doses, likely due to a more cautious approach in prescribing for patients with CKD whose renal clearance of drugs may be impaired. See my specific comments for the authors in the attached pdf file. In general ,this article may benefit from discussing more details regarding different types of pain in CKD, as well as use of non-opioid pharmacological agents and non-pharmacological strategies for pain management. Of particular interest is the potential role of dietary phosphorus restriction, often recommended in CKD, which should be investigated in future studies of pain management, especially for pain associated with CKD-bone mineral disorders.

We look forward to receiving your revised manuscript.

Kind regards,

Lalit Gupta

Academic Editor

PLOS One

Journal Requirements:

Additional Editor Comments :

Clinical guidelines are lacking for opioid use in patients with chronic kidney disease (CKD), who have higher risks for overdose and mortality from high-dose opioid use. The authors' study found that patient data stratified by eGFR generally showed a lower risk of escalation to higher opioid doses, likely due to a more cautious approach in prescribing for patients with CKD whose renal clearance of drugs may be impaired. See my specific comments for the authors in the attached pdf file. In general ,this article may benefit from discussing more details regarding different types of pain in CKD, as well as use of non-opioid pharmacological agents and non-pharmacological strategies for pain management. Of particular interest is the potential role of dietary phosphorus restriction, often recommended in CKD, which should be investigated in future studies of pain management, especially for pain associated with CKD-bone mineral disorders.

Reviewers' comments:

Reviewer's Responses to Questions

**Comments to the Author**

1. Is the manuscript technically sound, and do the data support the conclusions?

Reviewer #1: Yes

Reviewer #2: Yes

2. Has the statistical analysis been performed appropriately and rigorously?

Reviewer #1: Yes

Reviewer #2: Yes

3. Have the authors made all data underlying the findings in their manuscript fully available?

Reviewer #1: Yes

Reviewer #2: Yes

4. Is the manuscript presented in an intelligible fashion and written in standard English?

Reviewer #1: Yes

Reviewer #2: Yes

Reviewer #1: Clinical guidelines are lacking for opioid use in patients with chronic kidney disease (CKD), who have higher risks for overdose and mortality from high-dose opioid use. The authors' study found that patient data stratified by eGFR generally showed a lower risk of escalation to higher opioid doses, likely due to a more cautious approach in prescribing for patients with CKD whose renal clearance of drugs may be impaired. See my specific comments for the authors in the attached pdf file. In general ,this article may benefit from discussing more details regarding different types of pain in CKD, as well as use of non-opioid pharmacological agents and non-pharmacological strategies for pain management. Of particular interest is the potential role of dietary phosphorus restriction, often recommended in CKD, which should be investigated in future studies of pain management, especially for pain associated with CKD-bone mineral disorders.

Reviewer #2: This is a well-conducted and clearly presented study addressing an important and clinically relevant question. The methodology is robust, the statistical analysis is appropriate, and the findings are interpreted in a balanced and thoughtful manner. I have no further comments and recommend acceptance in its current form.

**Do you want your identity to be public for this peer review?** For information about this choice, including consent withdrawal, please see our Privacy Policy

Reviewer #1: No

Reviewer #2: **Yes:** Dr. Devang Bharti

---

## [Author Response · Author response to Decision Letter 1]

27 Feb 2026

24 February 2026

Dear Academic Editor

PLOS ONE

Revision of Manuscript Entitled: Patterns of opioid dose escalation in patients with chronic kidney disease initiated on opioids for the treatment of non-cancer pain (PONE-D-25-59036)

We would like to sincerely thank you for your careful review of our manuscript and for your constructive and insightful comments. We are grateful for the positive evaluation of our work and for the valuable suggestions, which have helped us to improve the clarity, clinical relevance, and overall quality of the manuscript.

We have carefully revised the manuscript in response to all comments. All changes have been highlighted in the revised version using track changes. Below, we provide a point-by-point response to the issues raised.

Response to Editor’s Comments

Comment:

The manuscript would benefit from further discussion on different types of pain in CKD, non-opioid pharmacological agents, non-pharmacological strategies, and the potential role of dietary phosphorus restriction.

Response:

We have expanded the Discussion section to include a more detailed description of pain phenotypes in CKD, alternative pharmacological therapies, non-pharmacological interventions, and nutritional considerations related to CKD–mineral and bone disorders. We have also highlighted the limited empirical evidence regarding dietary phosphorus restriction and framed this as an important direction for future research. These additions are presented in the revised Discussion section (pages 332-348).

Response to Reviewer #1

Comment:

The article may benefit from discussing different types of pain in CKD, non-opioid pharmacological agents, non-pharmacological strategies, and dietary phosphorus restriction.

Response:

Same as above comments (pages 332-348)

Data Availability Statement

Comment:

Please clarify restrictions on data sharing and provide appropriate access information.

Response:

We have revised the Data Availability Statement as below:

“ The data used in this study originate from Swedish national healthcare registers and are protected under the General Data Protection Regulation (GDPR) and Swedish legislation. Therefore, the data cannot be shared publicly. Researchers interested in accessing Swedish register data may submit an application to the Swedish National Board of Health and Welfare (Socialstyrelsen) in accordance with Swedish law. Information on how to apply for data access is available at: https://www.socialstyrelsen.se/en/statistics-and-data/apply-for-data/ or by contacting registerservice@socialstyrelsen.se.”

We believe that these revisions have strengthened the manuscript and addressed all concerns raised by the Editor and reviewers. We sincerely appreciate the time and effort invested in reviewing our work and thank you for the opportunity to revise and improve the manuscript. We hope that the revised version will now be suitable for publication in PLOS ONE.

Thank you for your kind consideration.

Yours sincerely,

Che Suraya Zin, PhD

Department of Pharmacy Practice

Kulliyyah of Pharmacy, International Islamic University Malaysia

Email: chesuraya@iium.edu.my

Juan-Jesús Carrero, PhD

Department of Medical Epidemiology and Biostatistics

Karolinska Institutet, Stockholm, Sweden

Email: juan.jesus.carrero@ki.se

---

## [Decision Letter · Decision Letter 1]

5 Mar 2026

Patterns of opioid dose escalation in patients with chronic kidney disease initiated on opioids for the treatment of non-cancer pain

PONE-D-25-59036R1

Dear Dr. Carrero,

We’re pleased to inform you that your manuscript has been judged scientifically suitable for publication and will be formally accepted for publication once it meets all outstanding technical requirements.

Kind regards,

Lalit Gupta

Academic Editor

PLOS One

Additional Editor Comments (optional):

The manuscript has been carefully revised in accordance with the thoughtful recommendations provided by the reviewers in the previous round. All concerns have been thoroughly addressed, resulting in what we believe is a significantly improved and more rigorous paper. This manuscript is now in an acceptable form and meets the high standards required for publication.

Reviewers' comments:

Reviewer's Responses to Questions

**Comments to the Author**

Reviewer #1: All comments have been addressed

2. Is the manuscript technically sound, and do the data support the conclusions?

Reviewer #1: Yes

3. Has the statistical analysis been performed appropriately and rigorously?

Reviewer #1: Yes

4. Have the authors made all data underlying the findings in their manuscript fully available?

Reviewer #1: Yes

5. Is the manuscript presented in an intelligible fashion and written in standard English?

Reviewer #1: Yes

Reviewer #1: I am satisified with the authors' revision regarding alternative pharmacological, non-pharmacological, and other therapies for pain management, including nutritional guidance recommendatios, for different types of pain in chronic kidney disease.

**Do you want your identity to be public for this peer review?** For information about this choice, including consent withdrawal, please see our Privacy Policy

Reviewer #1: No

---

## [Editor Report · Acceptance letter]

PONE-D-25-59036R1

PLOS One

Dear Dr. Carrero,

I'm pleased to inform you that your manuscript has been deemed suitable for publication in PLOS One. Congratulations! Your manuscript is now being handed over to our production team.

Kind regards,

on behalf of

Dr. Lalit Gupta

Academic Editor

PLOS One